# Construction of Highly Active Zn_3_In_2_S_6_ (110)/g-C_3_N_4_ System by Low Temperature Solvothermal for Efficient Degradation of Tetracycline under Visible Light

**DOI:** 10.3390/ijms232113221

**Published:** 2022-10-30

**Authors:** Haohao Huo, Yuzhen Li, Shaojie Wang, Siyang Tan, Xin Li, Siyuan Yi, Lizhen Gao

**Affiliations:** 1College of Environmental Science and Engineering, Taiyuan University of Technology, 79 Yingze Street, Wanbailin District, Taiyuan 030024, China; 2School of Mechanical Engineering, University of Western Australia, 35 Stirling Highway, Perth, WA 6009, Australia

**Keywords:** Zn_3_In_2_S_6_/g-C_3_N_4_, tetracycline, low temperature, photocatalytic

## Abstract

Herein, Zn_3_In_2_S_6_ photocatalyst with (110) exposed facet was prepared by low temperature solvothermal method. On this basis, a highly efficient binary Zn_3_In_2_S_6_/g-C_3_N_4_ was obtained by low temperature solvothermal method and applied to the degradation of tetracycline (TC). The samples of the preparation were characterized by X-ray diffraction, scanning electron microscope, transmission electron microscope, UV–vis diffuse reflection spectroscopy, and photoluminescence spectroscopy. Furthermore, the degradation performance of photocatalysts on TC was investigated under different experimental conditions. Finally, the mechanism of Zn_3_In_2_S_6_/g-C_3_N_4_ composite material degrading TC is discussed. The results show that Zn_3_In_2_S_6_ and Zn_3_In_2_S_6_/g-C_3_N_4_ photocatalysts with excellent performance could be successfully prepared at lower temperature. The Zn_3_In_2_S_6_/g-C_3_N_4_ heterojunction photocatalyst could significantly improve the photocatalytic activity compared with g-C_3_N_4_. After 150 min of illumination, the efficiency of 80%Zn_3_In_2_S_6_/g-C_3_N_4_ to degrade TC was 1.35 times that of g-C_3_N_4_. The improvement of photocatalytic activity was due to the formation of Zn_3_In_2_S_6_/g-C_3_N_4_ heterojunction, which promoted the transfer of photogenerated electron–holes. The cycle experiment test confirmed that Zn_3_In_2_S_6_/g-C_3_N_4_ composite material had excellent stability. The free radical capture experiment showed that ·O_2_^−^ was the primary active material. This study provides a new strategy for the preparation of photocatalysts with excellent performance at low temperature.

## 1. Introduction

The development of antibiotics brings convenience to people, but also pollutes the ecological environment [1,2]. Residual antibiotics in the environment destroy the activity of microorganisms and the balance of the ecosystem, which will affect the survival and growth of animals and plants, and then threaten human beings. In the past few decades, various methods have been studied and used in the treatment of antibiotic wastewater, such as the adsorption method [3,4,5], membrane separation method [6], Fenton oxidation method [7,8], biological method [9], and photocatalytic method [10,11,12]. Among them, the photocatalytic method has become a popular way for antibiotic wastewater treatment due to its high efficiency, environmental protection, and characteristics that can be carried out under normal temperature and pressure. Photocatalysis is an advanced oxidation technology with the semiconductor photocatalyst as the core. Its main principle is that semiconductors produce potent oxidizing substances under visible light irradiation, then these substances react with pollutants in water and ultimately degrade pollutants. Photocatalysis technology can deal with increasingly serious environmental pollution through the efficient use of solar energy.

Graphite carbon nitride (g-C_3_N_4_) is a medium bandgap semiconductor (2.6~2.7 eV) with pale yellow layered structure, which has good visible light response [13,14,15]. The central structural units of g-C_3_N_4_ are triazine ring and heptaazine ring. In these two units, the carbon atoms and nitrogen atoms are sp^2^ hybridized to form a π-π conjugated structure, similar to two-dimensional graphene, and then stacked to form a three-dimensional crystal structure [16,17,18]. Since Wang and Markus discovered that g-C_3_N_4_ has the catalytic performance of visible light to catalyze water splitting with the help of the cocatalysts and sacrificial agents, g-C_3_N_4_ has received widespread attention [19]. G-C_3_N_4_ not only has excellent photoelectric properties and easily adjustable electronic structure, but also has excellent thermochemical stability and good biocompatibility [20,21]. Therefore, g-C_3_N_4_ is popular in the field of photocatalysis. However, as with other single-component photocatalysts, the high photogenerated charge recombination rate and low specific surface area limit its practical application [22,23,24]. At present, many methods have been proposed to adjust and improve the photocatalytic performance of g-C_3_N_4_, such as morphology control [25,26,27], element doping [28,29,30], and construction of heterostructures [31,32,33,34,35]. The construction of type II heterojunction between g-C_3_N_4_ and other semiconductors shows higher desirability because the staggered energy level arrangement of the two semiconductors promotes the separation of photogenerated electron–hole pairs, thereby improving the photocatalytic performance [36].

Recently, ternary chalcogen compounds have been widely used in the field of photocatalysis because of their suitable bandgap energy, unique electronic structure, and optical properties of tunable wavelength. Zn_3_In_2_S_6_ is a ternary chalcogenide compound with a bandgap energy of 2.8 eV, which has suitable conduction and valence band positions, and sufficient redox potential [37]. Zn_3_In_2_S_6_ has shown great potential in the photocatalytic degradation of pollutants and hydrogen evolution. However, the practical application of Zn_3_In_2_S_6_ is limited by the unstable and slow photogenerated electron–hole pair separation [38,39,40]. At present, many heterojunction composite materials based on Zn_3_In_2_S_6_ have been reported, such as Polypyrrole/Zn_3_In_2_S_6_ [40], NiS/Zn_3_In_2_S_6_ [37], and Ni_2_P/Zn_3_In_2_S_6_ [41]. However, there are few studies on using Zn_3_In_2_S_6_ as a cocatalyst. According to the energy band matching mechanism, it is possible to construct a heterojunction between g-C_3_N_4_ and Zn_3_In_2_S_6_. This is mainly because the conduction band of g-C_3_N_4_ (−1.15 eV vs. NHE) is more negative than that of Zn_3_In_2_S_6_ (−0.90 eV vs. NHE), which provides the possibility for the transfer of photogenerated electrons. In addition, since the valence band of Zn_3_In_2_S_6_ (+1.90 eV vs. NHE) is more positive than that of g-C_3_N_4_ (+1.49 eV vs. NHE), the h^+^ in the valence band of Zn_3_In_2_S_6_ will be transferred to the valence band of g-C_3_N_4_, thus achieving efficient separation of photogenerated electron–hole pairs. In previous studies, Zn_3_In_2_S_6_ had been combined with g-C_3_N_4_ to further enhance the photocatalytic activity, such as Zn_3_In_2_S_6_/FCN [39], g-C_3_N_4_/Zn_3_In_2_S_6_ [42], and NiS/Zn_3_In_2_S_6_/g-C_3_N_4_ [43]. However, the high temperature reaction process in the hydrothermal method increases the cost and safety risk of catalyst preparation. In addition, Shi et al. [44] prepared ZnIn_2_S_4_ nanosheets exposing (110) facet to provide more active sites at low temperature by adding surfactants, which improved the photocatalytic activity of ZnIn_2_S_4_. Inspired by this, we want to know whether the Zn_3_In_2_S_6_ photocatalyst with more exposed active facet can be prepared at low temperature. 

In this work, the Zn_3_In_2_S_6_/g-C_3_N_4_ photocatalyst was prepared by the low temperature solvothermal method, and its photocatalytic activity for tetracycline (TC) under visible light irradiation was evaluated. The microscopic morphology and photoelectric properties of the Zn_3_In_2_S_6_/g-C_3_N_4_ photocatalyst were analyzed by XRD, SEM, TEM, UV–Vis diffuse, and PL. The optimal doping ratio of Zn_3_In_2_S_6_ nanosphere was determined through the experiments. Further, on the premise of the optimal doping ratio, the photocatalytic performance of the catalyst was studied under different experimental parameters (catalyst dosage, initial concentration, and initial pH value of the solution). In addition, the stability of the Zn_3_In_2_S_6_/g-C_3_N_4_ photocatalyst was studied through cycling experiments. Finally, through the free radical capture experiment, the active material that plays a major role in the reaction process was determined, and the possible photocatalytic mechanism of Zn_3_In_2_S_6_/g-C_3_N_4_ was discussed.

## 2. Results and Discussion

### 2.1. Characterization Analyses of Samples

Figure 1 shows the XRD patterns of the prepared g-C_3_N_4_, Zn_3_In_2_S_6_, and 80ZIS/CN. As shown in Figure 1, in the XRD pattern of g-C_3_N_4_, there are two prominent diffraction peaks at 27.40° and 13.04°, which are respectively attributed to the (002) crystal plane and (001) crystal plane (JCPDS 87-1526). The weak diffraction peak at 13.04° belongs to the periodic three-s-triazine ring structure, while the strong diffraction peak at 27.40° is a typical interlayer stacking of conjugated aromatics [45]. In the XRD pattern of Zn_3_In_2_S_6_, the diffraction peaks at 27.15°, 28.43°, 47.17°, and 56.00° correspond to the (011), (102), (110), and (022) lattice planes, respectively (JCPDS 89-3964). The diffraction peak of 80 ZIS/CN at 27.40° corresponds to the (002) crystal plane of g-C_3_N_4_, and the diffraction peak here is obviously broadened compared with g-C_3_N_4_ and Zn_3_In_2_S_6_. The reason for this phenomenon is that the diffraction peak of Zn_3_In_2_S_6_ at 27.15° coincides with that of flake g-C_3_N_4_ at 27.40°. In addition, the diffraction peaks at 47.17°, 56.00°, and 76.17° of 80 ZIS/CN correspond to the (110), (022), and (212) crystal planes of Zn_3_In_2_S_6_, which indicates that Zn_3_In_2_S_6_ is successfully loaded on the g-C_3_N_4_. In addition, no additional peaks are found on 80 ZIS/CN, indicating that the sample with high purity is prepared. 

The morphology and microstructure of the prepared photocatalyst observed by the SEM and TEM, and the results are shown in Figure 2. G-C_3_N_4_ exhibits a pleated laminar structure, which allows them to have a large specific surface area for better reaction with contaminants (Figure 2a). Zn_3_In_2_S_6_ nanosphere presents microspherical appearances of different sizes and the surface of these microspheres is composed of a large number of petal pieces, so the unique structure should have a higher specific surface area and more abundant surface active sites (Figure 2b). In addition, it can be seen that the microsphere-shaped Zn_3_In_2_S_6_ and the flake-shaped g-C_3_N_4_ overlap each other and combine with each other, indicating that Zn_3_In_2_S_6_ is successfully loaded on the g-C_3_N_4_ (Figure 2c). Figure 2d shows that many Zn_3_In_2_S_6_ nano-particles (darker black dots in the Figure 2d) are dispersed on the thin g-C_3_N_4_ nanosheets (the g-C_3_N_4_ nanosheets show a laminar structure, Figure 2a), indicating that Zn_3_In_2_S_6_ is successfully loaded on g-C_3_N_4_. Figure 2e is a TEM photograph of 80ZIS/CN at high magnification, which can further show that the lattice fringe with a crystal interplanar spacing of 0.31 nm corresponds to the (102) crystal plane of Zn_3_In_2_S_6_. In addition, an obvious heterojunction interface can be observed, which further indicates that Zn_3_In_2_S_6_ and g-C_3_N_4_ form a heterojunction. The elemental composition of the binary 80ZIS/CN composite was further determined by EDS. As shown in Figure 2f, there are C, N, In, S, and Zn elements in the 80ZIS/CN composite, which are uniformly distributed, indicating that Zn_3_In_2_S_6_ is successfully loaded the on g-C_3_N_4_.

The light absorption characteristics of the materials are different, and the degree of response to sunlight is also different. The optical absorption characteristics and energy band structure of g-C_3_N_4_ nanosheet, Zn_3_In_2_S_6_ nanosphere, and binary 80ZIS/CN photocatalytic materials were analyzed by UV–Vis DRS, and the results are shown in Figure 3. The optical absorption edge of g-C_3_N_4_ is about 463 nm. For Zn_3_In_2_S_6_ nanosphere, the edge source of absorption light is located at 481 nm. The absorption edge of the binary 80ZIS/CN heterojunction formed by doping Zn_3_In_2_S_6_ on g-C_3_N_4_ is about 473 nm. Compared with g-C_3_N_4_, the absorption edge of 80ZIS/CN is red-shifted, indicating that 80ZIS/CN has a stronger light absorption capacity and shows higher light utilization than g-C_3_N_4_. This phenomenon may be caused by the interaction between g-C_3_N_4_ and Zn_3_In_2_S_6_, which improves the separation efficiency of photogenerated electron–hole pairs [46]. 

The bandgap energy (*E_g_*) of g-C_3_N_4_ nanosheet, Zn_3_In_2_S_6_ nanosphere, and binary 80ZIS/CN photocatalysts can be calculated by the following formula [47]:
*ahv* = *A* (*hv* − *E_g_*)*^n^*^/2^(1)
where: *E_g_*, *v*, *h*, *A* and *a* are bandgap energy, optical frequency, planck’s constant, proportionality constant and the absorption coefficient, respectively. 

According to previous literature, g-C_3_N_4_ is a direct bandgap semiconductor [48], while Zn_3_In_2_S_6_ is an indirect bandgap semiconductor [49]. Therefore, the *n* value of g-C_3_N_4_ is 1 [48], and the *n* value of Zn_3_In_2_S_6_ is 4 [49]. The curves of *(ahv)^n/^*^2^ with respect to *hv* (eV) of g-C_3_N_4_, Zn_3_In_2_S_6_, and binary 80ZIS/CN are shown in Figure 3b. The *E_g_* values can be determined by extrapolating the tangent of the curve to the intercept of the *hv* axis. The *E_g_* values of g-C_3_N_4_, Zn_3_In_2_S_6_, and binary 80ZIS/CN photocatalysts are estimated to be approximately 2.64, 2.80, and 2.74 eV, respectively. 

The conduction band (*CB*) and valence band (*VB*) of the photocatalyst can be calculated by the following formula [46]:*E_VB_* = *X* − *E^C^* + 0.5*E_g_*(2)
*E_CB_* = *E_VB_* − *E_g_*(3)
where, *E_VB_* indicates the valence potential, eV; *X* denotes the electronegativity; *E^C^* indicates the energy of the free electrons on the hydrogen atom, about 4.5 eV; *E_g_* represents band gap energy of the sample, eV. Through the query, the *X* value of g-C_3_N_4_ is 4.67 eV [50]. After calculation, the *VB* potential of g-C_3_N_4_ is 1.49 eV. The *CB* potential of g-C_3_N_4_ can be calculated by *E_CB_* = *E_VB_* − *E_g_*, and its conduction band potential is −1.15 eV. The electronegativity of Zn_3_In_2_S_6_ is 5 eV [51]. Therefore, by calculation, the valence band potential of Zn_3_In_2_S_6_ is 1.90 eV. The conduction band edge potential (*E_CB_*) can be calculated by *E_CB_* = *E_VB_*
*− Eg*, and the conduction band potential of Zn_3_In_2_S_6_ is −0.90 eV. 

Fluorescence spectroscopy (PL) is an effective method to evaluate the efficiency of charge separation. The PL spectra of g-C_3_N_4_, Zn_3_In_2_S_6_ and binary 80ZIS/CN heterojunction are shown in Figure 4. Generally speaking, the lower the intensity of the peaks, the better the photoinduced electron–hole separation effect, and vice versa, the worse the effect. It can be observed from Figure 4 that g-C_3_N_4_ exhibits a strong fluorescence emission peak, which is due to its high photogenerated electron–hole recombination rate. The PL emission peak of Zn_3_In_2_S_6_ at the same position is almost negligible, which may be because the number of photogenerated electron–hole pairs generated is lower than that of g-C_3_N_4_ under the same illumination [52]. The luminescence intensity of 80ZIS/CN at the same position is significantly lower than that of g-C_3_N_4_, indicating that the photogenerated electron hole pairs in the complex have been effectively separated relative to g-C_3_N_4_.

The chemical bonds and functional groups of the prepared catalysts were analyzed by FT-IR spectra. As shown in Figure 5, the results show that 80ZIS/CN displays feature similar to g-C_3_N_4_. The broad peak at 3000~3400 cm^−1^ is caused by stretching vibration of N-H bond and O-H bond. The peak in the range of 1200~1700 cm^−1^ is attributed to the stretching vibration of the C-N heterocyclic chemical bond. Two sharp peaks at 812 cm^−1^ and 880 cm^−1^ point to the planar skeleton bending vibration of tri-s-triazine and triazine units in g-C_3_N_4_. Among them, peaks at 1620 and 1394 cm^−1^ were found on 80ZIS/CN, attributed to surface-bound water molecules and hydroxyl groups in Zn_3_In_2_S_6_ [45]. According to the FT-IR results, the binary ZIS/CN composite was successfully prepared.

Figure 6 shows the EIS spectra of g-C_3_N_4_, Zn_3_In_2_S_6_, and 80ZIS/CN. From the diagram, 80ZIS/CN shows the smallest radius of curvature, indicating that the introduction of Zn_3_In_2_S_6_ is beneficial to reduce the charge transfer resistance, thereby improving the photocatalytic activity. The results show that the combination of Zn_3_In_2_S_6_ and g-C_3_N_4_ effectively prolongs the lifetime of photogenerated electron–hole pairs.

### 2.2. Photocatalytic Activity of Samples

Figure 7 show the degradation effect of g-C_3_N_4_, Zn_3_In_2_S_6_, and x-ZIS/CN binary composites on TC solution under visible light irradiation and the degradation rate constant of the photocatalyst for TC. As shown in Figure 7a, before adding the catalyst, no TC is degraded, indicating that the TC is stable at room temperature. Before illumination, the adsorption performance of the prepared photocatalyst was tested in a dark environment, and the reaction time was 40 min. The results show that g-C_3_N_4_ exhibits the lowest dark adsorption capacity for TC, and only 5.9% of TC is adsorbed within 40 min. In contrast, Zn_3_In_2_S_6_ reveals the best performance, that 21.7% of TC is adsorbed within 40 min. For the x-ZIS/CN composite, with the increase in Zn_3_In_2_S_6_ doping ratio, the amount of dark adsorption gradually increases and is all higher than g-C_3_N_4_. When the doping ratio of Zn_3_In_2_S_6_ increases from 20% to 100%, the dark adsorption amount increases from 9.5% to 18.4%. Specifically, when the doping ratio of Zn_3_In_2_S_6_ is 20%, 40%, 60%, 80%, and 100%, respectively, the amount of dark adsorption at this time is 9.5%, 15.3%, 13.7%, 15.1%, and 18.4%, respectively, which indicate that the doping of Zn_3_In_2_S_6_ is beneficial to improve the dark adsorption capacity of g-C_3_N_4_. In the light reaction, when the doping ratio of Zn_3_In_2_S_6_ increases from 20% to 80%, the degradation rate of TC increases from 78.7% to 85.7% after 150 min. In particular, the degradation efficiency of TC is 78.7%, 81%, 84%, and 85.7%, when the dosing ratio of Zn_3_In_2_S_6_ is 20%, 40%, 60%, and 80%, respectively. However, when the doping ratio is further increased to 100%, the degradation effect of TC reveals a downward trend at this time, only 82.4%, which is 3.3% lower than that of 80% of Zn_3_In_2_S_6_. The reasons for this trend can be attributed to: (1) the increase in Zn_3_In_2_S_6_ doping ratio leads to the increase in heterojunctions number, which inhibits the recombination of photogenerated electron–hole pairs and promotes the photocatalytic activity [53]; (2) however, the high doping ratio of Zn_3_In_2_S_6_ will cause agglomeration of Zn_3_In_2_S_6_ and therefore the number of effective heterojunctions and photocatalytic activity is reduced [54]. Therefore, based on the above discussion, 80ZIS/CN was selected as the best photocatalyst for the following experiments.

Figure 7b shows the variation of TC degradation rate constant of different prepared samples. In Figure 5b, g-C_3_N_4_ exhibits a lower degradation rate constant than the x-ZIS/CN complex, only 0.00709 min^−1^. However, for x-ZIS/CN, when the doping ratio of Zn_3_In_2_S_6_ is 20%, 40%, 60%, 80%, and 100%, the degradation rate constants is 0.01422 min^−1^, 0.01581 min^−1^, 0.01773 min^−1^, 0.02156 min^−1^, 0.01961 min^−1^, indicating that the doping of Zn_3_In_2_S_6_ promotes the performance of the g-C_3_N_4_ photocatalyst. Furthermore, for the x-ZIS/CN composite, the rate constant increases first and then decreases with the increase in doping rate. The 80ZIS/CN has the best doping ratio of 0.02155 min^−1^, which is consistent with the results shown in Figure 7a. Therefore, 80% is the optimal doping ratio in this study.

The dosage of photocatalyst has an essential influence on the degradation effect of TC in water. The photocatalytic degradation experiment was carried out to determine the optimal dosage of 80ZIS/CN, and the results are shown in the Figure 8. Figure 8a,b show the variation of the photocatalytic degradation of TC with different dosages of 80ZIS/CN and the photodegradation rate constants at 40 min, respectively.

As shown in Figure 8a, as the dosage of 80ZIS/CN binary catalyst increases, the adsorption capacity of TC also increases in the dark reaction stage. When the dosage of catalyst is 0.200, 0.240, 0.320, 0.400 and 0.480 g/L, the adsorption capacity of 80ZIS/CN binary catalyst for TC is 15.1%, 23%, 27.5%, 30.6%, and 38.2%, respectively. The increase in the photocatalyst adsorption capacity may be attributed to the increase in surfactant sites, which in turn improves the adsorption capacity for TC. After 120 min of illumination, when the dosage of 80ZIS/CN increases from 0.200 g/L to 0.320 g/L, the photodegradation efficiency of TC increases from 82.7% to 84.5%. However, when the dosage of 80ZIS/CN further increases to 0.480 g/L, the photodegradation efficiency drops to 84%. The results show that appropriate amount of photocatalyst can promote the degradation of TC, and the excessive amount of photocatalyst will have the opposite effect. The reason for this result may be that as the dosage increases, the number of active sites in the solution increases, which promotes the degradation of TC; however, too much catalyst leads to the increase in solution turbidity, which reduces the light transmittance of the solution, and eventually leads to the decrease in photocatalytic efficiency [55].

As shown in Figure 8b, after 40 min of illumination, the degradation rate of TC increases firstly and then decreases with the increase in the dosage. When the dosage is 0.320 g/L, the photodegradation rate constant is the maximum of 0.02799 min^−1^, corresponding to Figure 8a. Therefore, 0.320 g/L was selected as the optimal dosage for the next experiment.

In practical application, the concentration of pollutants is not a fixed value. TC solutions with different initial concentrations were selected as the target pollutants to evaluate the degradation performance of the 80ZIS/CN photocatalyst. The results are exhibited in Figure 9. As Figure 9a shows, the adsorption capacity of the 80ZIS/CN photocatalyst for TC gradually decreases with the solution concentration increases in the dark reaction stage. Among them, when the concentration of TC solution increases from 10 mg/L to 30 mg/L, the dark adsorption of TC decreases from 43.1% to 21.5%. Specifically, when the concentration of TC solution is 10, 15, 20, 25, and 30 mg/L, the dark adsorption of TC is 43.1%, 34.4%, 27.5%, 23.9%, 21.5%, respectively. After illumination, with the increase in the initial concentration of the TC, the photocatalytic effect gradually reduces, from 89.3% to 78.0%. The reasons for this result are as follows: (a) the increase in pollutant concentration in the solution will increase the difficulty of light penetration in the solution and reduce the transmission path of photons, so as to reduce the chance of photons migrating to the surface of the photocatalyst, resulting in the reduction of photocatalytic efficiency [56,57]; (b) as the concentration of pollutants increases, the competition between TC molecules and intermediates in the degradation process for reaction sites will be intensified, thus affecting the photocatalytic effect [58].

Figure 9b exhibits the photodegradation rate constant of the photocatalyst at 40 min under different initial concentrations of the solution. In Figure 9b, the degradation rate of TC increases first and then decreases with the initial solution concentration. When the concentration of TC solution changes from 10 mg/L to 20 mg/L, the photodegradation rate constant increases from 0.02619 min^−1^ to 0.028 min^−1^, an increase of 6.91%. However, when the solution concentration is further increased to 30 mg/L, the degradation rate reduces to 0.01978 min^−1^. In the case of the solution concentration of 20 mg/L, the degradation rate is the fastest, which is 0.028 min^−1^. Therefore, 20 mg/L was used to the best initial solution concentration for the next experiments.

During the photocatalytic reaction process, the pH of the solution has an effect on the surface charge distribution of the catalyst and pollutants, and the oxidation potential of the catalyst valence band [59]. Therefore, it is imperative to carry out photocatalytic degradation experiments under different pH conditions. Figure 10 shows the process profile of TC degradation by photocatalyst at different pH values and the UV–Vis spectra of binary 80ZIS/CN photodegradation of TC at different times. In Figure 10a, it is clear that the dark adsorption capacity of TC molecule increases with the increase in pH value. In the photoreaction stage, the degradation efficiency of TC shows a trend of first increasing and decreasing with the increase in pH value. After 120 min of light, as the pH value increases from 3.02 to 7.01, the degradation rate of TC increases from 70.3% to 82.5%, while the pH increases to 11.0, 68.7% of TC molecules were degraded. When the pH is 7.01, the best degradation rate of TC is 82.5%. There are three main reasons believed for this phenomenon. The pH of the solution has an essential effect on the surface charge distribution of the catalyst and pollutants. When the solution is too acidic or too alkaline, the surface of the catalyst and the TC are both positively and negatively charged. When the charges are the same, they repel each other, which affects the adsorption of TC molecules, and further affecting the photocatalytic performance [59]. When the pH of the solution increases, the mutual repulsion between the photocatalyst and the pollutants changes into electrostatic attraction, which leads to an increase in adsorption capacity. However, the excessively high electrostatic attraction will reduce reaction sites on the catalyst surface, reduce the production of active species, and reduce the photocatalytic activity. The pH of the solution determines the adhesion mode between the pollutants and the photocatalyst, so that the pollutants have different degradation pathways during the reaction, thus exhibiting different photocatalytic effects [58,60,61]. As shown in Figure 8a, when the pH is 7.01, the photocatalyst has the highest photocatalytic degradation effect, and 82.5% of the TC could be degraded within 60 min. Therefore, follow-up experiments were carried out at this pH value.

Figure 10b shows the UV–visible spectrum of the TC solution under optimal conditions. From Figure 10b, it can be observed that a typical absorption peak of TC solution exists at 357 nm, and the intensity of the peak at the maximum absorption wavelength gradually decreases as the photocatalytic reaction proceeds, and almost disappears completely at 120 min illumination.

The reusability of the prepared 80ZIS/CN photocatalyst was tested, and the results are shown in Figure 11. It can be observed from Figure 11 that the degradation rates of 20 mg/L TC by 80ZIS/CN are 82.44%, 82.44%, 80.92%, and 79.40% after the four cycles of the test, respectively, and the final degradation rate decreases by about 3.04%, which may be attributed to the loss of samples during multiple centrifugation processes [62]. However, the overall decline of photocatalyst efficiency has a slow downward trend, and the rate of decline is low, which indicated that the photocatalyst still exhibits relatively stable physical and chemical properties after multiple cycles and exhibits a better degradation cycle effect.

The ·OH, h^+^ and ·O_2_^−^ produced during the experiment were captured by adding TBA of 0.5 mmol/L, AO of 0.05 mmol/L, and BQ of 0.0126 mg/L into the solution. The result is shown in Figure 12a. When TBA and AO were added to the solution, although the degradation effect of TC is slightly lower than that of the blank, the degradation effect of TC is not significantly reduced after 120 min of light, so h^+^ and ·OH are not the main active species. When BQ was added, the degradation effect of TC is 42.75% after 120 min illumination, which is significantly reduced compared with the blank. Thus, in the experimental process of 80ZIS/CN nanocomposite material for photocatalytic degradation of TC, ·O_2_^−^ is the active group that plays a major role.

In the reaction between NBT and ·O_2_^−^, when NBT is excessive, only monoformazan is produced in the system; when ·O_2_^−^ is excessive, NBT is first reduced to produce monoformazan, and monoformazan continues to be reduced to produce diformazan. The fluorescent probe experiment is shown in Figure 12b. Weighed 50 mg of catalyst was added to 250 mL of distilled water and then ultrasonically dispersed for 10 min, then 10.2205 mg of NBT was added, and the photocatalytic reaction was carried out in the photocatalytic reactor. During the photoreaction process, 10 mL of the solution was taken at 5 min, 10 min, 20 min, 30 min, 60 min, 90 min, and 120 min, and then centrifuged three times at 10,000 rpm for 5 min each time, and the absorbance was measured on an ultraviolet spectrophotometer. During the reaction, ·O_2_^−^ can react with NBT to reduce it, resulting in a decrease in the intensity of NBT with the increase in light time, and the intensity decreased after 120 min, indicating the formation of ·O_2_^−^ on the catalyst surface, which in turn counter-proves that the active factor playing a major role in photocatalysis is ·O_2_^−^.

Based on the results of the capture experiment, the possible electron transfer paths of the photocatalytic degradation of TC by the 80ZIS/CN binary photocatalyst could be clarified, and the mechanism diagram is shown in Figure 13. Both Zn_3_In_2_S_6_ and g-C_3_N_4_ are excited under visible light, e^-^ transfers from the VB to the CB, and h^+^ is left on the conduction band. Because the CB potential of g-C_3_N_4_ (−1.15 eV/NHE) is more negative than the standard oxidation–reduction potential of O_2_/·O_2_^−^ (−0.33 eV/NHE), the photogenerated electrons on the CB of g-C_3_N_4_ will interact with oxygen to generate superoxide radicals (·O_2_^−^), thereby degrading TC. In addition, the VB (+1.49 eV) of g-C_3_N_4_ is more negative than Zn_3_In_2_S_6_ (+1.9 eV), and the CB (−1.15eV) of g-C_3_N_4_ is more negative than Zn_3_In_2_S_6_ (−0.9 eV), so the electrons in the g-C_3_N_4_ conduction band are going to go to the Zn_3_In_2_S_6_ conduction band, and the holes on the VB of Zn_3_In_2_S_6_ will move to the VB of g-C_3_N_4_ under light irradiation, which promotes the spatial separation of electron–hole pairs, thereby reducing the recombination of electron–hole pairs and prolonging the lifetime of photogenerated electron–hole pairs. As a result, a large number of electrons on the CB of Zn_3_In_2_S_6_ combine with O_2_ in the water to generate a large amount of ·O_2_^−^, which then degrades TC. Since the VB energy of Zn_3_In_2_S_6_ (+1.9 eV/NHE) and g-C_3_N_4_ (+1.53 eV/NHE) is lower than ·OH/H_2_O (+1.99 eV/NHE) and ·OH/OH^−^ (+ 2.40 eV/NHE), the photogenerated holes on the valence band cannot react with H_2_O or OH^−^ to generate ·OH, which further confirms that the hydroxyl radical (·OH) is not the main active group during the experiment. In addition, according to the capture experiment, it can be known that the h^+^ on the VB of g-C_3_N_4_ does not play a significant role in the entire photocatalytic degradation process, so it is not the main active species. In sum, in the process of photocatalytic reaction, the leading group is ·O_2_^−^.

## 3. Materials and Methods

### 3.1. Chemicals

Urea (H_2_NCONH_2_) chemicals were obtained from Damao Chemical Reagent Factory (Tianjin, China). Indium chloride (InCl_3_·4H_2_O) was obtained from the Aladdin reagent. Absolute ethanol (CH_3_CH_2_OH), ammonium oxalate (AO), thioacetamide (C_2_H_5_NS), and ethylene glycol (HOCH_2_CH_2_OH) were obtained by Zhiyuan Chemical Reagent Co., Ltd. (Tianjin, China). Hydrochloric acid (HCl) was obtained from Feng Chuan Chemical Reagent Technology Co., Ltd.(Tianjin, China), and so was sodium hydroxide (NaOH). Tert-butyl alcohol (TBA) and zinc chloride (ZnCl_2_) were applied from Beichen Fang zheng Reagent Factory (Tianjin, China). TC (C_22_H_24_N_2_O_8_) chemicals were purchased from Sinopharm Chemical Reagent Co., Ltd. (China). P-benzoquinone (BQ) was obtained from Qinghua Jinying Technology Co., Ltd. (Tianjin, China). Sodium citrate (C_6_H_5_Na_3_O_7_·2H_2_O) was got from Kaitong Chemical Reagent Co., Ltd. (Tianjin, China). Nitro Tetrazolium Blue (NBT) was purchased from BASF Biotechnology Co., Ltd. (Hefei, China). During the experiment, All the chemicals had the characteristics of analytical purity and did not need secondary purification. Ultrapure water was used in the experiment. 

### 3.2. Synthesis of Zn_3_In_2_S_6_/g-C_3_N_4_ Heterojunctions

#### 3.2.1. Preparation of g-C_3_N_4_ Nanosheet

The g-C_3_N_4_ nanosheet was prepared by the thermal polymerization method [63]. The detailed steps are as follows. The 20 g precursor urea was placed into alumina oxide crucible with lid, heat to 824.15 K at a heating rate of 279.15 K/min in the electric furnace, and keep it for 240 min. After the reaction process is completed, the yellow powder cooled to room temperature was ground and collected. A certain amount of the above massive yellow powder was added to the covered alumina crucible, and raise to 794.15 K at a heating rate of 279.15 K/min in the electric furnace for 120 min. After the reaction process was completed, the g-C_3_N_4_ nanosheet with lighter color was obtained. Then the sample was labeled CN.

#### 3.2.2. Preparation of Zn_3_In_2_S_6_ Nanosphere

Zn_3_In_2_S_6_ nanosphere was obtained by the solvothermal method. Firstly, a certain amount of ZnCl_2_, InCl_3_·4H_2_O, and C_2_H_5_NS was weighed according to the chemical molar ratio (Zn:In:S = 3:2:6). Then the equal molar weight of C_6_H_5_Na_3_O_7_·2H_2_O with ZnCl_2_ was weighed. The weighed ZnCl_2_, InCl_3_·4H_2_O, and C_6_H_5_Na_3_O_7_·2H_2_O were dissolved into a mixed solution containing 10 mL of ethylene glycol and 50 mL of deionized water and ultrasonically dispersed for 30 min at room temperature. Secondly, the C_2_H_5_NS was added to the above system and stirred at room temperature for 30 min. Thirdly, the obtained solution was placed in the polytetrafluoroethylene liner and reacted at 394.15 K for 720 min. The obtained light yellow product was washed three times with distilled water and absolute ethanol respectively, and finally dried in an oven at 334.15 K overnight to obtain light yellow Zn_3_In_2_S_6_ powder, which was named after ZIS. The sample preparation flow chart is shown in Figure 1.

#### 3.2.3. Preparation of Zn_3_In_2_S_6_/g-C_3_N_4_ Heterojunction

The Zn_3_In_2_S_6_/g-C_3_N_4_ heterojunction was obtained by the solvothermal method. Firstly, a certain amount of ZnCl_2_, InCl_3_·4H_2_O, and C_6_H_5_Na_3_O_7_·2H_2_O were added into a solution containing 50 mL deionized water and 10 mL ethylene glycol at a stoichiometric ratio of 3:2:3, and the solution was ultrasonically dispersed for 30 min. Then, the 300 mg weighed CN was ultrasonically dispersed into the above solution and magnetically stirred for 30 min at room temperature. Moreover, adding TAA to the dispersion and stir the mixture at room temperature for 30 min. Finally, the obtained dispersion was poured into the autoclave and maintained at 394.15 K for 720 min. The obtained product was washed three times with distilled water and absolute ethanol. Then the obtained precipitate was dispersed in a petri dish containing absolute ethanol and dried in an oven at 334.15 K overnight. Zn_3_In_2_S_6_/g-C_3_N_4_ heterojunction was obtained. By adjusting the quality of Zn_3_In_2_S_6_, the Zn_3_In_2_S_6_/g-C_3_N_4_ complex with loadings of 20%, 40%, 60%, 80%, and 100% can be obtained and labeled as 20ZIS/CN, 40ZIS/CN, 60ZIS/CN, 80ZIS/CN, and 100ZIS/CN. The sample preparation flow chart is shown in Figure 1.

### 3.3. Characterization

TD-3500 X-ray diffractometer (XRD) for determination of crystal structure of samples. The measurement conditions were using Cu-Kα as a source (λ = 0.154 nm), a scanning speed of 5 °/min, voltage of 40 kV, and current of 30 mA. The morphology and microstructure of the samples were obtained using field emission scanning electron microscopy (FE-SEM) (JSM-7100F) and transmission electron microscopy (TEM) (JEM-2010) at 200 kV. The composition and content of elements in the sample were determined by energy dispersive X-ray energy spectrometer (EDS) on scanning electron microscope. The UV–Vis diffuse reflectance spectrum was obtained on a Hitachi U-3900 U–Vis spectrophotometer. The instrument uses barium sulfate as reference material. Shimadzu RF-6000 fluorescence spectrophotometer obtained photoluminescence (PL) spectra of samples. The Zeta potential measurement was measured as follows: 5 mg of g-C_3_N_4_, Zn_3_In_2_S_6_, and Zn_3_In_2_S_6_/g-C_3_N_4_ nanocomposite was added to 10 mL of distilled water and then sonicated for 15 min (40 kHz, 600 W bath type sonicator), respectively. Further, 1 mL of suspension was injected into the cell for the determination of zeta potential (Nano ZS90 Malvern, UK).

### 3.4. Photocatalytic Performance Evaluation

TC was selected as the degradation object to test the catalytic performance of the prepared samples. During the experiment, a 300 W Xenon lamp with a 420 nm cut-off filter was used as the light source and a 300 mL quartz beaker as the reactor. In the reaction process, the quartz beaker was placed on the magnetic stirrer to ensure the uniform mixing of the solution during the reaction. In addition, the outside of the beaker was surrounded by an ice water mixture to ensure that the temperature remained constant during the reaction. First, the photocatalyst sample was ultrasonic dissolved into 250 mL TC solution (C_TC_ = 20 mg/L). The dark reaction was carried out for 40 min under dark conditions to reach adsorption saturation. Then turned on the Xenon lamp, and the solution was taken every other period of time, then centrifuged 2 times. The supernatant was filtered with 0.45 μm filter membrane. The supernatant was filtered with 0.45 μm filter membrane. Finally, the filtrate was tested by UV-1800PC ultraviolet–visible spectrophotometer (China). The following equation was used to calculate the photocatalytic efficiency (*η*) of TC:*η* = C/C_0_(4)
where, C_0_ is the initial concentration of TC solution, C is the instantaneous concentration of TC solution in the solution at time *t* (min). The following formula was used to calculate the first-order rate constant in the photocatalytic process:k_obs_ = (−1/*t*) ∗ ln(C/C_0_)(5)
where, k_obs_ represents the photodegradation rate constant (min^−1^), the TC concentration after 40 min dark adsorption (mg/L) is C_0_, C represents the instantaneous concentration of the TC solution at irradiation time *t* (min). 

### 3.5. Free Radical Capture Experiments

After the dark reaction, TBA, BQ, and AO were added to the TC solution (C_TC_ = 10 mg/L, pH = 7.01) to explain the effect of capture hydroxyl radicals (·OH), superoxide radicals (·O_2_^−^) and holes (h^+^) on TC degradation. Then the remaining process was similar to the photocatalytic process.

## 4. Conclusions

In this paper, Zn_3_In_2_S_6_ nanosphere with (110) exposing facet was prepared by the low temperature solvothermal method at a low temperature, which showed excellent degradation performance for TC. In addition, the binary Zn_3_In_2_S_6_/g-C_3_N_4_ heterojunction was prepared by the solvothermal method, and showed excellent photocatalytic performance for TC. The degradation effects of binary photocatalysts on TC were obtained under different conditions (doping ratio of Zn_3_In_2_S_6_, catalyst dosage, initial concentration of the solution, initial pH value of the solution), and the degradation mechanism of photocatalysts on TC was analyzed. Based on the experimental results, the following conclusions can be drawn:

(1) The Zn_3_In_2_S_6_ and Zn_3_In_2_S_6_/g-C_3_N_4_ photocatalysts with excellent performance could be successfully prepared at a lower temperature.

(2) The characterization results show that the loading of Zn_3_In_2_S_6_ reduces the photogenerated electron–hole pair complexation rate of g-C_3_N_4_ and improves its photocatalytic activity. 

(3) Degradation experiments show that, compared with g-C_3_N_4_, x-Zn_3_In_2_S_6_/g-C_3_N_4_ composite exhibits excellent degradation performance, and 80ZIS/CN has the best degradation effect. After 120 min of visible light irradiation, 0.320 g/L 80ZIS/CN could degrade 84% of the 20 mg/L TC. 

(4) Through the free radical capture experiment, it is confirmed that the ·O_2_^−^ had an important effect on the degradation of TC. Binary Zn_3_In_2_S_6_/g-C_3_N_4_ composites show higher photochemical stability after four cycles, and it is further known that the type II electron transfer mechanism formed between g-C_3_N_4_ and Zn_3_In_2_S_6._

The binary Zn_3_In_2_S_6_/g-C_3_N_4_ composite prepared in this study expands the way of pollutant treatment in wastewater.

## Data Availability

Not applicable.

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
