# Peer review of "Construction of Highly Active Zn3In2S6 (110)/g-C3N4 System by Low Temperature Solvothermal for Efficient Degradation of Tetracycline under Visible Light"

_ijms, 2022, doi:10.3390/ijms232113221_

Round 1

Reviewer 1 Report

In this paper, the authors conducted a research about the heterojunction formation of g-C3N4/Zn3In2S6 for tetracycline degradation prepared by low temperature solvothermal reaction. 

Although the title is looks interesting, however, the content of this manuscript has to be polished significantly. There are some major point that authors have to be considered.

1. The Surface area analysis has to be conduccted to understand the adsorption behavior of the sample.

2. The phootocatalytic result from the composites are not as high as Zn3In2S6 sample, therefore, we doubt that the heterojunction formation was formed. 

3. from the XRD result the author said "Moreover, there are no other impurity peaks, indicating the prepared monomers all have high crystallnity" This statement is confusing because the impurity and crystallinity are both not directly correlated.

4. From the UV-VIS result, the Tauc-plot analysis of Zn3In2S6 is quite strange, is the calculation is correct?

5. The terminology of "doped" is not suitable in this case because this system is a heterojunction system.

Author Response

Ms. Ref. No.:  #IJMS-1950614

Manuscript Title: Construction of highly active Zn3In2S6 (110)/g-C3N4 system by low temperature solvothermal for efficient degradation of tetracycline under visible light

Dear Ms. Karolina Janković,

Thank you for your letter and the reviewers' valuable suggestions for our manuscript. Thank you very much for your help in our article. We are ready to respond in detail to the comments made by reviewers and editors. According to the proposal, we revised it and marked it in RED in the second revision. The detailed corrections are listed below point by point :

Reviewer 1: In this paper, the authors conducted a research about the heterojunction formation of g-C3N4/Zn3In2S6 for tetracycline degradation prepared by low temperature solvothermal reaction. Although the title is looks interesting, however, the content of this manuscript has to be polished significantly. There are some major point that authors have to be considered.

Question 1-1:The Surface area analysis has to be conduccted to understand the adsorption behavior of the sample. 

Response 1-1: Thank you very much for reviewer’s suggestion. As you said, the adsorption behavior of the sample in the photocatalytic reaction is an important factor that cannot be avoided. After receiving your valuable opinions, we should have sent the samples at the first time. Unfortunately, due to the epidemic, we were unable to complete the surface analysis of the sample within the specified time, which will be an irreparable shortcoming in our experiments. In addition, according to many previous published studies [1-5] and the analysis of the structure of the paper, although the surface analysis of the sample plays an important role in understanding the photocatalytic behavior of the sample, combined with the catalytic mechanism and experimental design structure analysis, effective electron transfer occupies a more important position in this paper. Based on the existing experimental conditions, we further improved the FT-IR and EIS analysis to further analyze the structure and electron transfer of the complex. Again, deep apologies for the inability to complete the surface analysis of the sample.

  1. Qiu, P.; Yao, J.; Chen, H.; Jiang, F.; Xie, X. Enhanced Visible-Light Photocatalytic Decomposition of 2,4-Dichlorophenoxyacetic Acid over ZnIn2S4/g-C3N4 J. Hazard. Mater.2016, 317, 158–168. 
  2. Luan, J.; Chen, M.; Hu, W. Synthesis, Characterization and Photocatalytic Activity of New Photocatalyst ZnBiSbO4under Visible Light Irradiation. Int. J. Mol. Sci. 2014, 15, 9459–9480.
  3. Vinesh, V.; Preeyanghaa, M.; Kumar, T.R.N.; Ashokkumar, M.; Bianchi, C.L.; Neppolian, B. Revealing the Stability of CuWO4/g-C3N4Nanocomposite for Photocatalytic Tetracycline Degradation from the Aqueous Environment and DFT Analysis. Environ. Res. 2022, 207, 112112.
  4. Wu, C.; Zuo, H.; Zhang, S.; Zhao, S.; Du, H.; Yan, Q. A Novel Strategy to Construct a Direct Z-Scheme Bi@Bi2O2CO3/g-C3N4Heterojunction Catalyst via PDA Electronic Bridge. Sep. Purif. Technol. 2022, 294, 121242.
  5. Soltani, N.; Saion, E.; Hussein, M.Z.; Erfani, M.; Abedini, A.; Bahmanrokh, G.; Navasery, M.; Vaziri, P. Visible Light-Induced Degradation of Methylene Blue in the Presence of Photocatalytic ZnS and CdS Nanoparticles. Int. J. Mol. Sci. 2012, 13, 12242–12258.

Question 1-2:The phootocatalytic result from the composites are not as high as Zn3In2S6 sample, therefore, we doubt that the heterojunction formation was formed.

Response 1-2: Thank you very much for reviewer’s advise. From the XRD pattern of Figure 1, the characteristic peaks belonging to g-C3N4 and Zn3In2S6 can be found in the 80ZIS/CN complex. In addition, from the TEM map of Figure 2e, the (102) crystal plane belonging to Zn3In2S6 can be observed and the C, In, N, S, Zn elements can be found in the EDS map of Figure 2f, which further confirms the formation of heterojunction. The FT-IR spectra (Figure 5) of 80ZIS / CN further confirmed the formation of heterojunction. As can be seen from Figure 3a, Zn3In2S6 has a red shift relative to 80ZIS/CN, indicating that it has a stronger response capability under visible light, which may be the reason for its high reactivity.

Figure 1. XRD patterns of g-C3N4 nanosheets, Zn3In2S6 and binary 80ZIS/CN heterostructure

Figure 2e. TEM images of 80ZIS/CN

Figure 2f. SEM images of (a) g-C3N4, (b) Zn3In2S6, (c) 80ZIS/CN, (d), (e) TEM images of 80ZIS/CN, and (f) EDS elements mapping of 80ZIS/CN

Figure 5. PL spectra: g-C3N4, Zn3In2S6 and 80ZIS/CN heterojunction

Figure 3a. UV–vis spectra (b) the plots of (αhv)n/2 versus hv: g-C3N4 nanosheets, Zn3In2S6 and binary 80ZIS/CN.

Question 1-3: from the XRD result the author said "Moreover, there are no other impurity peaks, indicating the prepared monomers all have high crystallnity" This statement is confusing because the impurity and crystallinity are both not directly correlated.

Response 1-3: Thank you very much for reviewer’s advise. The presence of impurity peaks in the XRD pattern is not necessarily related to the crystallinity of the sample. The crystallinity of the sample can be obtained by fitting analysis. The "Moreover, there are no other impurity peaks, indicating the prepared monomers all have high crystallnity" has been deleted from the revised manuscript. Thank you very much for your valuable comments, we will be more careful consideration of each sentence scientific research terms, enhance the level of scientific research. 

Question 1-4: From the UV-VIS result, the Tauc-plot analysis of Zn3In2S6 is quite strange, is the calculation is correct?

Response 1-4: Thank you very much for reviewer’s advise. According to your opinion, we have tested the optical absorption properties of Zn3In2S6 samples and recalculated them. According to the study of Duan et al. [6], in the process of Zn preparation, the different sulfur sources in the precursor will have a great influence on the light absorption characteristics and band gap energy of Zn3In2S6 . The sulfur source used in this paper is thioacetamide, and the band gap energy of the obtained sample is similar to that of literature [6,7].

  1. Duan, S.; Zhang, S.; Chang, S.; Meng, S.; Fan, Y.; Zheng, X.; Chen, S. Efficient Photocatalytic Hydrogen Production from Formic Acid on Inexpensive and Stable Phosphide/Zn3In2S6Composite Photocatalysts under Mild Conditions. Int. J. Hydrog. Energy. 2019, 44, 21803–21820.
  2. Meng, S.; Wu, H.; Cui, Y.; Zheng, X.; Wang, H.; Chen, S.; Wang, Y.; Fu, X. One-Step Synthesis of 2D/2D-3D NiS/Zn3In2S6Hierarchical Structure toward Solar-to-Chemical Energy Transformation of Biomass-Relevant Alcohols. Appl. Catal. B-Environ. 2020, 266, 118617.

Question 1-5: The terminology of "doped" is not suitable in this case because this system is a heterojunction system.

Response 1-5: Thank you very much for reviewer’s comment. We re-examine the usage of "doped", which is inappropriate for heterojunctions. We replace "doped" with "loaded" and correct it in the revised manuscript.

              In line 120 of page 3 of revised draft, "Zn3In2S6 is successfully doped on the g-C3N4" is replaced by "Zn3In2S6 is successfully loaded on the g-C3N4".

              In line 132 of page 4 of revised draft, "indicating that Zn3In2S6 is successfully doped on the g-C3N4" is replaced by "indicating that Zn3In2S6 is successfully loaded on the g-C3N4".

              In line 143 of page 4 of revised draft, "indicating that Zn3In2S6 is successfully doped on the g-C3N4" is replaced by "indicating that Zn3In2S6 is successfully loaded on the g-C3N4".   

We acknowledge the reviewer’s comments and suggestions very much, which are valuable in improving the quality of our manuscript.

We sincerely hope that the revised manuscript is now acceptable for publication in the journal “Internation journal of molecular sciences”. We look forward to hearing from you soon.
   If there are other errors or further requests, please contact me by e-mail.

With best regards.

Yours sincerely,

Dr. Yuzhen Li (Y.-Z. Li)

College of Environmental Science and Engineering

Taiyuan University of Technology

79 Yingze Street

Yingze District, Taiyuan, 030024

China

Email: liyuzhen123456@126.com,liyuzhen@tyut.edu.cn

Author Response

Ms. Ref. No.:  #IJMS-1950614

Manuscript Title: Construction of highly active Zn3In2S6 (110)/g-C3N4 system by low temperature solvothermal for efficient degradation of tetracycline under visible light

Dear Ms. Karolina Janković,

Thank you for your letter and the reviewers' valuable suggestions for our manuscript. Thank you very much for your help in our article. We are ready to respond in detail to the comments made by reviewers and editors. According to the proposal, we revised it and marked it in RED in the second revision. The detailed corrections are listed below point by point :

Reviewer 3: It is an original study and the original aspect of the study is strong. The results have been interpreted successfully. The manuscript was written in appropriate way. The parts of this manuscript are adequately and clearly expressed. Especially, the introduction part is well explained. The literature was studied in detail and summarized in the manuscript. The data are robust enought to draw the

conclusions. The results and the conclusions are interesting for the readership of this journal and will attract a wide readership depending on the scope of this journal. As a result, the work can be accepted published without any revision. By the way, the english language of this manuscript is appropriate and understandable.

Response: Thank you reviewers for your recognition.Thank you again for your contribution to this manuscript.

We acknowledge the reviewer’s comments and suggestions very much, which are valuable in improving the quality of our manuscript.

We sincerely hope that the revised manuscript is now acceptable for publication in the journal “Internation journal of molecular sciences”. We look forward to hearing from you soon.
   If there are other errors or further requests, please contact me by e-mail.

With best regards.

Yours sincerely,

Dr. Yuzhen Li (Y.-Z. Li)

College of Environmental Science and Engineering

Taiyuan University of Technology

79 Yingze Street

Yingze District, Taiyuan, 030024

China

Email: liyuzhen123456@126.com,liyuzhen@tyut.edu.cn

Reviewer 3 Report

In this manuscript the synthesis of Zn3In2S6/g-C3N4 composites with a low temperature solvothermal method is described. This material is properly characterized, and its use in the degradation of TC in investigated. 

From my point of view, it has the quality to be published in IJMS, but not before mejor revision.

Some suggestions:

Page 2, line 79 - what does "corrected" mean?

Page 2, line 93 - UV should be changed by UV diffuse reflectance

The characterization could be improved with XPS, FTIR and EIS

Page 3 - You should explain why you characterize only 80ZIS/CN. It is explained in photo degradation experiments, but not in this moment

Figure 2e is not clear enough to support what you state in page 4, lines 133-136

Page 5, line 154: Planck's constant

Page 6, line 176: spectra instead of spectras

Page 7: doping ratio of Zn3In2S6 in composites is not clear. What does 100% doping ratio? It is not equivalent to pure Zn3In2S6.

Page 8: the values for dosage of 80ZIS/CN are not justified 

Page 9: the values for initial concentration of TC solution are not justified

The results in photo degradation experiments could be improved by measuring TOC (mineralization) or even the nature and concentration of degradation products by chromatography  

Page 12: I do not understand the "saliva" term in this context 

Round 2

Reviewer 1 Report

The manuscript can be accepted in this formed.

Reviewer 3 Report

The manuscript could be published in the present form.